# A Systematic Review of Endothelial Dysfunction in Chronic Venous Disease—Inflammation, Oxidative Stress, and Shear Stress

**DOI:** 10.3390/ijms26083660

**Published:** 2025-04-12

**Authors:** Hristo Abrashev, Despina Abrasheva, Nadelin Nikolov, Julian Ananiev, Ekaterina Georgieva

**Affiliations:** 1Department of Vascular Surgery, Medical Faculty, Trakia University, 6000 Stara Zagora, Bulgaria; hristo.abrashev@trakia-uni.bg; 2II Department of Internal Medicine Therapy: Cardiology, Rheumatology, Hematology and Gastroenterology, Medical Faculty, Trakia University, 6000 Stara Zagora, Bulgaria; despina.abrasheva@trakia-uni.bg; 3Vascular Surgery Department, National Heart Hospital, 1000 Sofia, Bulgaria; nadelin.nikolov@gmail.com; 4Department of General and Clinical Pathology, Forensic Medicine, Deontology and Dermatovenerology, Medical Faculty, Trakia University, 6000 Stara Zagora, Bulgaria; julian.r.ananiev@trakia-uni.bg

**Keywords:** chronic venous disease, CVD, inflammation, endothelial dysfunction, oxidative stress, reactive oxygen species, shear stress

## Abstract

Chronic venous disease (CVD) is among the most common diseases in industrialized countries and has a significant socioeconomic impact. The diversity of clinical symptoms and manifestations of CVD pose major challenges in routine diagnosis and treatment. Despite the high prevalence and the huge number of venous surgical interventions performed every day, a substantial proportion of the etiopathogenesis remains unclear. There are several widely advocated and generally valid theories of “peri-capillary fibrin cuffs” and “white cell trapping hypothesis”, which consider the role of venous reflux/obstruction, inflammation, vascular remodeling, hemodynamic changes, genetic and social risk factors. There are several specific provoking factors for the development of venous reflux: incompetence of the valve system, inflammation of the vascular wall, and venous hypertension. Over the past few years, increasing scientific data has demonstrated the link between oxidative stress, endothelial dysfunction, and vascular inflammation. High levels of oxidants and persistent inflammation can cause cumulative changes in hemodynamics, resulting in permanent and irreversible damage to the microcirculation and endothelial cells. Production of reactive oxygen species and expression of inflammatory cytokines and adhesion molecules are involved in a vicious cycle of venous wall remodeling. The interaction of ROS, and in particular, the superoxide anion radical, with nitric oxide leads to a decrease in NO bioavailability, followed by the initiation of prolonged vasoconstriction and hypoxia and impairment of vascular tone. This review addresses the role of ED, oxidative, and hemodynamic stress in the CVD mediation. Based on predefined inclusion and exclusion criteria, we conducted a systematic review of published scientific articles using PubMed, PMC Europe, Scopus, WoS, MEDLINE, and Google Scholar databases in the interval from 24 April 2002 to 1 April 2025. The current review included studies (*n* = 197) scientific articles, including new reviews, updates, and grey literature, which were evaluated according to eligibility criteria. The selection process was performed using a standardized form according to PRISMA rules, the manual search of the databases, and a double-check to ensure transparent and complete reporting of reviews. Studies had to report quantitative assessments of the relationship between vascular endothelial dysfunction, inflammation, oxidative stress, and shear stress in a chronic venous disease.

## 1. Introduction

Cardiovascular diseases are multifactorial disorders with high prevalence in the population and are among the leading cause of death worldwide [1,2,3]. Chronic venous disease (CVD) is characterized by specific signs and clinical symptoms (most often affecting the lower extremities) that are due to impaired venous drainage and associated with venous hypertension. It can be defined as any morphological and functional abnormality of the venous system of long duration that presents with symptoms and/or signs that require investigation and/or further treatment [4]. Over the past 30 years, several significant randomized clinical trials have been conducted on CVD prevalence. Analysis shows substantial heterogeneity and reports a worldwide prevalence of CVD that varies widely from 14.3% to over 83.6% [5,6,7].

Numerous risk factors play a substantial role in the development and CVD progression [8,9]. Among the most prominent are age, gender, ethnicity, family history (for CVD and previous venous thromboembolism), obesity (BMI > 30 kg/m^2^), heavy physical labor (more than 5 h per day in a standing or sitting position), past pregnancies (number of childbirths), hormone replacement therapy and contraception, menopause, smoking, genetic diseases, systemic diseases and congenital connective tissue diseases, flat feet, and immobility [10].

Normal endothelial function is a crucial factor for maintaining vascular homeostasis. The vascular endothelium has a key role in the maintenance of the metabolism of the vascular wall, while simultaneously performing the paracrine, endocrine, and autocrine function [11,12]. Under the influence of a number of humoral and hemodynamic stimuli, endothelial cells synthesize a large spectrum of mediators that regulate vascular tone, cell adhesion and proliferation of vascular smooth muscle cells (VSMCs), and inflammation of the venous wall [13,14]. Endothelial cells (ECs) are actively involved in the regulation and modulation of vascular permeability, angiogenesis, inflammatory and immune cell response (leukocyte migration and activation), hemostasis (fibrinolysis, thrombosis, and platelet activation) [15]. Endothelial cells are essential for the transport of hormones, peptides, lipoproteins, and other macromolecules [16].

Endothelial function is perhaps one of the most informative and early prognostic markers that is used for assessing the state of the cardiovascular system. Endothelial dysfunction is considered as a clinical syndrome that is associated with an increased rate of adverse cardiovascular events and imminent vascular wall damage [17,18]. Excessive secretion of adhesion and proinflammatory molecules by the endothelium stimulates reactive oxygen species (ROS) production and can lead to redox imbalance and endothelial activation [19]. The ROS generation in CVD leads to the uncoupling of the endothelial nitric oxide synthase (eNOS) and promotes progression in the endothelial dysfunction. Nitric oxide (NO) depletion is due to its interaction with superoxide anion radical (O_2_^●−^) and the formation of peroxynitrite (OONO^−^), which leads to further damage to endothelial cells and seriously impairs vascular homeostasis [20,21,22,23,24,25]. Excess ROS and reduced NO production, followed by ONOO^−^ formation, promote nitration of multiple proteins [26,27] and vascular remodeling impairing vasodilatory capacity [28]. This further initiates a new inflammatory response and procoagulation and contributes to mitochondrial damage, endothelial dysfunction, and endothelial cell death [29].

The vascular endothelium is in constant contact with blood flow. Various mechanical factors, such as changes in physiological blood pressure and shear stress (SS), continuously deform the ECs, causing continuous autoregulation of vascular tone and adequate tissue perfusion with changes in vessel diameter [30]. Recent clinical studies showed that hemodynamic stress is one of the main catalysts of the processes of endothelial cell phenotyping, as well as a cause of morphological damage and functional incompetence of the endothelium [31]. Our proposed systematic review presents the data of pathophysiological mechanisms involved in the development and progression of CVD, focusing entirely on the inflammation, oxidative, and shear stress in the vascular endothelium.

## 2. Methods

In clinical practice and in vascular medicine, chronic venous disease is considered primarily as a function of venous hypertension and valvular incompetence. Chronic inflammation and impaired redox homeostasis are the critical drivers of endothelial changes in the vascular wall, and the basic cause of disease complications and progression. The present review aimed to remind us of the biochemical mechanisms in endothelial injury and directed our attention to bridging the gap between the fundamental science and clinical practice in CVD.

Research questions that were addressed include the evidence for vascular endothelial dysfunction as a function of inflammation, oxidative stress, NO bioavailability, and changes in hemodynamic shear stress in CVD. The topic and research question were identified, and the data analysis and preparation in this review followed the requirements for systematic reviews and meta-analysis according to the PRISMA-P guidelines protocols [32].

### 2.1. Literature Search and Data Collection Process

This systematic review focused on the role of inflammation, oxidative stress, and shear stress in chronic venous disease and prioritized studies that directly assessed noncompartmental dysfunction in humans. The previously known data on the topic were revised. Our strategy included keyword selection with an optimal search filter and inclusion and exclusion strategies according to the PRISMA rules for reporting systematic reviews [32]. The review protocol was published and registered with ID CRD420251023122 in the PROSPERO platform. Priority was given to data from the last 10 years due to the modern possibility of molecular medicine for monitoring and detailed diagnosis of CVD patients by measurement of inflammatory markers and oxidative stress, and of shear stress assessment.

The available databases of PubMed, PMC Europe, Scopus, Web of Science (WoS), MEDLINE, and Google Scholar were used. For the purpose of this review, over 1508 clinical, laboratory, and epidemiological studies and literature reviews describing various aspects of CVD were reviewed. In the present study, we selected 197 free full articles and clinical trials written in English directly relevant to our topic. The process of data analysis and study selection is presented in Figure 1. All identified and eligible studies were screened for duplicates and reviewed independently by two reviewers at each stage of the study. Discrepancies in data extraction were resolved through discussion to ensure consistency and accuracy in the data collection process (Table 1).

#### 2.1.1. Inclusion Criteria

The number of identified original articles and studies presented the role of free radical damage, oxidative stress, and endothelial dysfunction in vascular pathology. We revised original non-randomized studies that focused and presented endothelial dysfunction as the triad of inflammation–oxidative stress–shear stress in CVD patients (≥18 years of age), patients with clinically diagnosed chronic venous disease, studies using validated diagnostic methods such as duplex ultrasound, assessment of clinical symptoms and CVD stage, and comprehensive coverage of biomarkers that provide clinical relevance. To ensure consistency with current understanding of CVD pathophysiology, we included publications in the period from 24 April 2002 to 1 April 2025 in the following areas: 1/“Chronic venous disease”, 2/“CVD etiology, 3/“Risk factors”, 4/“ROS”, 5/“Oxidative stress”, 6/“Nitric oxide and RNS”, 7/“Endothelial dysfunction”, 8/“Vascular biology”, 9/“Shear stress”, 10/“Inflammation”, 11/“Vascular endothelial cells”, 12/“Vascular dysfunction” and a combination of them.

#### 2.1.2. Exclusion Criteria

This review article did not include letters, comments, preprints, letters to the editor, and non-peer-reviewed articles such as conference abstracts. Here, we did not review acute or mixed vascular pathologies associated with oxidative damage and those with significant comorbidities. Studies without measurable outcomes related to endothelial dysfunction, such as without direct or indirect measurements of inflammation and redox mechanisms, shear stress, and articles in severe arterial, autoimmune, infectious diseases, diabetes, cancer, etc., were also excluded. To ensure relevance and quality, the review did not include animal models. Scientific research with insufficient data and non-English articles was also not included.

## 3. Results

### 3.1. CVD Etiology and CEAP Classification

According to the internationally accepted Clinical–Etiology–Anatomy–Pathophysiology classification (CEAP), the etiology of CVD can be considered as: primary (Ep); secondary (Es): secondary-intravenous (Esi) and secondary-extravenous (Ese); congenital (Ec) and CVD with an unknown etiological factor (En) [33,34]. Primary CVD is most often due to various genetic and environmental risk factors [1]. It is characterized by a systemic progressive structural weakness of the venous wall (dilatation), vascular insufficiency, and venous reflux (venous hypertension) [35]. The risk factors included age, gender, ethnicity, family history, heavy physical labor, past pregnancies, menopause, hormone replacement therapy, immobility, and others [36,37,38,39]. Secondary CVD is divided into two subgroups: secondary-intravenous, where there is a damage to the venous wall and venous valves; and secondary-extravenous, where the venous wall and venous valves remain intact, while there is an ongoing a local or systemic pathological process such as dysfunction of the “muscle pump”, pregnancy, external compression by a tumor-like mass, increased central venous and intra-abdominal pressure, May–Tuner syndrome, trauma, etc. It is most often due to a previous pathological process, which led to the disruption of the structural and functional integrity of the venous system [33,40]. Statistically, the most common cause of secondary CVD is a history of previous deep vein thrombosis (DVT). DVT is characterized by aseptic inflammation involving the venous wall and perivenous adjacent structures, which can permanently impair the morphological and functional integrity of the deep venous system. Approximately one-third of all the DVT patients who have partially recovered (incomplete resolution of thrombotic masses into the deep system) will develop post-thrombotic syndrome (PTS) [41]. PTS is characterized by a broad set of pathophysiological mechanisms: inflammation, obstruction, venous reflux, vascular dysfunction, venous hypertension, and remodeling, which can result in severe venous dysfunction and subsequent chronic venous insufficiency [41,42]. Venous claudication is one of the most pronounced and frequent clinical manifestation of PTS, leading to severe disability and significant impairment of the quality of life of all the patients [43]. In general, it is defined by the patients as pain and stiffness in the lower limb during physical activity, which usually subsides after rest. According to Galanaud et al., the incidence of venous claudication varies widely between 5–45% [44]. Trophic changes (venous ulcers) of the lower extremities occur in between 5–10% of PTS patients [45]. Although rare, vascular trauma can be indicated as an etiological factor for the development of secondary CVD, with incidence varying between 1–2% in the general population [46]. Penetrating injuries have the most significant share in secondary CVD, due to their high incidence and severe postoperative complications such as arteriovenous fistulas, penetrating, DVT, etc. [41,47]. Congenital CVD accounts for a small percentage of all secondary CVD cases. Inherited syndromes involving single-gene mutations and/or chromosomal aberrations are involved in the pathogenesis. Among the most common inherited disorders that affect the venous system are Klippel–Trenaunay syndrome, Chuvash Polycythemia, Ehlers–Danilos’ syndrome, Lymphedema distichiasis syndrome, and Parkes Weber syndrome [40,48,49,50,51].

### 3.2. Structure of Venous Vessels

The structure of venous vessels is highly specialized and precisely organized into three anatomical compartments (superficial compartment, saphenous compartment, and deep compartment) so that the venous system can efficiently perform its vital functions in the body [52,53]. The histological structure of venous vessels is represented by: 1. Tunica intima—the innermost layer, made up of mainly ECs, which are limited by the lamina basalis, also called the extracellular matrix; 2. Tunica media—the middle layer, which is composed of smooth muscle cells (their transverse arrangement contributes to the change in the caliber of the vessel) and small elastic fibers; 3. Tunica adventitia—the outermost layer. Tunica adventitia contains the vasa vasorum and nervi vasorum, which are responsible for the blood supply and innervation of the vessel and consists of connective tissue and elastic fibers. This layer provides strength, support, and elasticity to the venous vessel [54,55,56]. The shape of the venous wall varies considerably, depending on the pressure and the volume of blood flow. When the pressure is low, the volume of circulating blood is small, and the anterior and posterior walls of the vessel lumen are almost contiguous in an elliptical shape. Venous vessels are easily adaptable to different blood pressures, and adequate adaptation of venous pressure is achieved with only small changes in the diameter of the vessel (from 5 mmHg to 25 mmHg), which outlines the large capacity of the venous system [41,57]. Compared to arteries, venous vessels are characterized by thin walls, large surface areas, and high elasticity. Due to the pull of gravity or some pathological conditions, blood can accumulate in vein vessels, which can increase the pressure and fluids in the interstitial space. The low pressure of venous blood flow leads to adaptive changes in the diameter of venous vessels, and may be accompanied by edema, varicose veins, chronic venous insufficiency, and other complications [58].

## 4. Morphology and Function of the Vascular Endothelium

### 4.1. Endothelial Cell Morphology

The endothelium is a heterogeneous monolayer of highly specialized endothelial cells (ECs), which is the first barrier to all the circulating blood components: blood cells, pathogens, macro and mircomolecules [53,59,60]. ECs originate from the splanchnopleural mesoderm, and under the action of specific proteins, the differentiation of mesodermal cells into angioblasts occurs, which the first precursors are blood cells and ECs [61]. In the blood vessel wall, ECs are tightly attached to a basement membrane, and they are arranged along the horizontal axis of the blood vessel, thereby reducing the mechanical frictional forces (shear stress (SS)) generated by the blood flow [62]. Depending on the localization, stages of maturation and formation, and accompanying pathophysiological processes, the thickness of the endothelial layer varies between 20–200 µm [63,64,65]. The integrity of this monolayer is maintained by a dynamic cytoskeleton and by the communication and signaling of individual cells with the extracellular matrix (ECM). The endothelium is made up of metabolically active and highly specialized cells that take part in many vital processes, such as the regulation and modulation of vascular permeability and vascular tone, angiogenesis, inflammatory and immune cell response (leukocyte migration and activation) [15], hemostasis (fibrinolysis, thrombosis, and platelet activation), synthesis of hormones, endopeptides, lipoproteins, amines, reactive oxygen species (ROS), angiogenesis, and metastasis, etc. [53,66,67,68]. Physiologically, venous and arterial ECs have some significant differences. One of the essential distinguishing hallmarks for these functional inconsistencies is in the type and number of intercellular junctions. For example, intercellular junctions in arterial vessels are denser and have a greater number of receptors expressing VEGF and the blood-clotting factor von Willebrand factor [69]. There are differences in gene expression of ephrin B2 in arterial vessels and ephrin B4 genes in venous vessels and also in the secretion of NO (venous ECs tend to have a higher production) [70,71,72]. The structural and functional diversity of ECs is due to intercellular connections that are specific to different tissues [73]. According to the function, ECs are distinguished into three different types [74] (Figure 2).

### 4.2. Functions of the Endothelium

The endothelium is a complex, dynamic, and autonomous structure that combines various functions (Figure 3) [12,76,77]. It is located at the border between circulating blood and tissues and plays a fundamental role in the building of blood and lymphatic vessels and in the lining of body cavities [78].

### 4.3. Endothelial Dysfunction in CVD

In recent years, scientific research on the pathophysiology of CVD has revealed some specific molecular mechanisms and cellular interactions that have established endothelial dysfunction as one of the major predisposing factors for the development of the disease [79,80,81,82,83,84,85]. The endothelium can be negatively affected by a variety of stress factors such as genetic mutations, environmental factors, changes in shear stress levels, low levels of NO and hypoxia, immunodeficiency states, chronic inflammation, increased levels of oxidative stress (OS), etc. [24,86,87]. Changes in the structural and morphological integrity of the endothelial layer predetermine the development of endothelial dysfunction, which can be considered as a combination of several pathological processes: increased synthesis of proinflammatory mediators and secretion of adhesion molecules, mechanical stresses in the endothelium and vascular remodeling, redox imbalance [31,88]. Disproportion in quantitative and qualitative disorders in the composition of the basement membrane (BM) inevitably leads to subsequent endothelial damage endothelial dysfunction (ED) [89]. These changes may occur simultaneously or occur separately through time. Nowadays, a growing amount of scientific data shows that ED may even precede the first clinical manifestations of most cardiovascular diseases, including CVD, and it can be used as a reliable prognostic marker for impaired vascular homeostasis [18,90,91].

#### 4.3.1. Role of Inflammation in Pathogenesis of CVD

Chronic inflammation is one of the main mechanisms leading to structural and functional endothelial damage [15]. As a result of increased venous pressure in the venous system, ECs activation occurs and initiation of mechanisms of ED. It consists of increased secretion of proinflammatory and procoagulant molecules, decreased shear stress, activation of ROS, venous dilation, leukocyte migration, and activation [92,93]. Activation of ECs stimulates the secretion of various proinflammatory mediators such as IL-1, IL-6, IL-17, IFN-γ, TNFα [94], chemokines, and growth factors (VGFE-A, PDGF-BB, angiotensin II, etc.), leading to vascular smooth muscle cell (VSMC) and extracellular matrix hypertrophy. Simultaneous autoactivation and hypertrophy of VSMC have a huge negative effect on both normal function and integrity of the venous system [95,96,97,98]. VSMC proliferation and dedifferentiation cause significant loss of flexibility and contractility of the venous wall, which inevitably impairs vascular tone by promoting venous vasodilation in patients with advanced ED [13]. VSMC-induced dysregulation in collagen synthesis is associated with reduced elasticity and increased vascular wall rigidity and has a major role in the progression of CVD [99]. In patients with advanced ED, ECs acquire morphological and functional features resembling those of mesenchymal cells. This process is accompanied by the expression of proinflammatory molecules such as hs-CRP, TNFα, IL-1β, IL-6, IL-8, interferon-inducible protein-10, macrophage inflammatory protein-1a, and monocyte chemoattractant protein 1 (MCP-1) [100,101]. MCP-1 is one of the key chemokines that regulate the migration and infiltration of monocytes and macrophages and has a major role in the inflammatory response of the vascular wall in patients with CVD [102]. Also, the endothelial cells initiated the release of procoagulant agents that stimulate intravascular thrombosis: plasminogen activator inhibitor 1 (PAI-1), von Willebrand factor, and factor VIII, which creates serious prerequisites for the development of venous thrombosis [103,104,105,106].

#### 4.3.2. Role of ROS

At low concentrations, free radicals (reactive oxygen and nitrogen species (RONS)) play an important physiological role in numerous biological processes such as cell signaling, regulation of redox homeostasis, and protection against pathogens [107,108,109]. Within reference range, RONS act as signaling molecules, while at higher concentrations they become toxic to living organisms. They are defined as redox-active molecules that include various oxygen-containing radicals O_2_^●−^, ^●^OH, alkoxy radicals (RO^●^), peroxyl radicals (ROO^●^), non-radical species (singlet oxygen, ozone and hydrogen peroxide (H_2_O_2_)), peroxynitrite (OONO^−^), and nitric oxide (NO) [91,110]. ROS have a pronounced bivalent nature, which on the one hand mediates numerous physiological events, and on the other hand, their excess causes structural damage to various important macromolecules by modifying specific amino acid residues, chain fragmentation, changing the electrical charge of amino acid residues, enzyme inactivation, etc. [111]. In high concentrations, ROS have a pronounced pathological effect. They can cause oxidative modification of various subcellular organelles such as mitochondria, which are the main source of mitochondrial ROS (mROS) in ECs. The exact mechanism involves the enzymatic generation of oxidants by xanthine oxidase, nitric oxide synthases (NOSs), mitochondrial monoamine oxidase, and NAPDH oxidase. In ECs, changes in the expression of endogenous enzymes induce increased production of oxidative molecules and oxidative (OS) and nitrosative (NS) stress [112,113,114]. Oxidative stress associated with the decompensation and compromise of natural antioxidant cellular systems result in an imbalance between the formation and elimination of ROS and RNS in favor of prooxidant processes in the body [115,116]. In the endothelium, ROS are generated by various mechanisms that involve the enzymes nicotinamide adenine dinucleotide phosphate (NADPH) oxidase, xanthine oxidase, lipoxygenase, or eNOS [117]. ROS production by NADPH oxidase enzymes has been found to be increased by a number of agonists such as angiotensin II, thrombin, PDGF, and TNFα [118,119]. ROS have a central role in tissue damage during episodes of reduced blood flow (ischemia) and subsequent restoration of flow (reperfusion). Reperfusion, in turn, initiates a surge in oxidant production, leading to cellular damage, inflammation, and impaired tissue repair [120,121]. Excessive ROS production and OS induces endothelial remodeling by initiating apoptosis, fibrosis, and hypertrophy [122,123,124].

#### 4.3.3. Role of NO

Nitric oxide (NO) is the most potent endogenous vasodilator in the human body, and it selectively maintains vascular homeostasis by acting as a platelet inhibitor. It participates in tissue oxygenation by controlling mitochondrial O_2_ consumption by inhibiting cytochrome C oxidase, regulating the modulation of inflammatory reactions, and the activation of the body’s immune response to pathogens [125,126,127]. It takes part in various physiological processes, including vasodilation, blood pressure regulation, inhibition of platelet aggregation, modulation of inflammation, mediation of cell signaling, and regulation of synaptic activity [128,129]. The interaction of NO with molecular oxygen or ROS (especially superoxide anion radical), generates the highly reactive radicals towards a number of intra- and extracellular targets, RNS and peroxynitrite (ONOO^−^) [130,131]. ECs are a rich source of vasoactive substances. ECs regulate the process of vasoconstriction and vasodilation by releasing vasoactive mediators such as NO, prostacyclin I2 (PGI2) and endothelial hyperpolarizing factor (EDHF), and natriuretic peptide (CNP), while vasoconstriction is mainly mediated by endothelin-1 (ET-1), angiotensin II (ACE2), thromboxane A2, and prostaglandin H2 [132]. ECs maintain low levels of OS and physiological levels of NO. Also, the cells provide reliable vascular protection, reducing OS, inhibiting VSMC migration and proliferation, and inhibiting leukocyte adhesion. Dysregulation of vasodilation and vasoconstriction, as well as increased levels of ROS, inflammatory mediators, and NO deficiency, promote phenotypic modulation of ECs and mediate ED [133,134]. Low levels of endogenous antioxidant systems and elevated levels of ROS cause inactivation and significantly reduce physiological NO bioavailability, which initiates the secretion of a number of proinflammatory mediators such as IL-8, IL-1, Il-6, IFN-γ, TNFα, and MCP-1 [135,136,137]. Furthermore, changes in the levels of endogenous NO and ROS-mediated endothelial damage significantly impair vasodilation, increase vascular permeability, and promote the secretion of proinflammatory mediators, leading to structural ECs changes–endothelial hypertrophy [21,24,91,138,139].

### 4.4. Basement Membrane—Structure and Function

The ECs’ surfaces are differentiated into two distinct regions: luminal and basolateral. The basolateral surface of ECs is isolated from surrounding tissues by a glycoprotein basement membrane. The luminal surface is the endothelial cell side, which is directed toward the bloodstream. The luminal membrane is involved in blood clotting regulation and immune cell adhesion and allows the exchange of molecules between the blood and the vessel wall [140,141]. The basement membrane maintains the integrity of the endothelial layer through certain adhesion molecules and mediators, ensuring the stable attachment of ECs. The firm attachment of ECs and the well-structured monolayer ensure laminar blood flow in the vessels and prevent platelet adhesion. Maintenance of physiological hemostasis is achieved by changes in the density and arrangement of collagen fibers in the basement membrane [60,142]. The basement membrane serves as a “solid foundation” for ECs adhesion. It is characterized by pronounced heterogeneity and tissue specificity in terms of composition and includes four main structural-determining components: collagen type IV, laminin, perlecan, and nidogen [143,144]. The BM’s heterogeneous structure and composition determine a rich set of functions, including selective transport of nutrients, hormones, specific proteins, and micro and macro molecules from the bloodstream to the connective tissue [145,146]. The basement membrane has a pronounced bipolar permeability, which allows for the deposition and utilization of metabolic products from tissues. It also regulates leukocyte reactivity in inflammation and synthesizes various growth factors and procoagulant molecules [147]. The basement membrane is directly involved in the angiogenesis, signaling, and tissue-regeneration processes by various peptides such as laminin deposition [148]. Chronic venous hypertension and venous stasis lead to mechanical stress on the endothelial cells and the basement membrane, which can cause thickening and fragmentation of the BM. Alterations in the composition and structure of the basement membrane lead to the stimulation of abnormal angiogenesis, disruption of the various signaling pathways and the intercellular junction, and increased vessel permeability. The loss of contact with the basement membrane causes leakage of fluid, proteins, and inflammatory cells into the surrounding tissues, severe inflammation, dysregulation of ECs, and endothelial dysfunction [81,144].

### 4.5. Shear Stress

Venous vessels are constantly exposed to two main types of hemodynamic forces—shear stress and peripheral stretch force—circumferential wall stress [149,150]. The term “shear stress” (SS) defines the frictional forces of the blood flow near the endothelial surface, and the peripheral stretching force represents the tension exerted by blood pressure on the entire circumference of the venous vessel [151,152,153]. The magnitude of SS is directly proportional to the velocity of blood flow in the vessel and the viscosity of the blood, and is inversely proportional to the radius of the blood vessel [154,155]. Increased venous pressure in the lower extremities directly initiates mechanical stimulation (mechanotransduction) of ECs and VSMCs, which results in the secretion of certain specific biochemical mediators that promote inflammation [84]. Mechanotransduction is a set of adaptive and regulatory processes in which cellular mechanosignaling occurs in response to various physical (mechanical) and hemodynamic stimuli (shear stress/circumferential wall stress), involving cells of the cardiovascular system (EC, VSMC, cardiomyocytes, leukocytes, erythrocytes, platelets), as well as some neurobaroreceptors [31,156]. The process has considerable time heterogeneity, alternating from rapid events (milliseconds to seconds) such as changes in membrane potential and intracellular calcium concentration, to medium-term events (minutes to hours) such as changes in cellular architecture, and long-term events (days to months) such as vascular remodeling [157].

Shear stress levels play an essential role in several processes: regulation of vascular wall elasticity and topography, expression of numerous anti-inflammatory, antithrombotic, and antioxidant mediators that stabilize the vascular wall, and control of vascular apoptosis [152,155,158]. Under physiological conditions, high shear stress has a strong protective effect against leukocyte adhesion [155], while increased levels of SS activate endothelial cells, initiating the secretion of NO and prostacyclin. This adaptive compensatory mechanism limits the negative effector mechanisms of inflammatory mediators with a powerful chemotaxis effect on neutrophils and macrophages [132,159,160,161]. Venous hypertension and dilatation significantly reduced SS levels in the endothelium while simultaneously increasing peripheral stretching forces. The low SS levels initiated biomolecular signaling, which triggers a “vicious cycle” in which increased venous pressure leads to venous wall remodeling and chronic inflammation [40,155]. During the inflammation, leukocytes bind to ECs by the location on the cell surface adhesion molecules ICAM, VCAM, laminin, E-, L-, and P-selectin), which plays the role of ligands and thus encourages adhesion of lymphocytes, monocytes, eosinophils, and basophils [162,163]. Increased levels of adhesion molecules lead to the accumulation of activated polymorphonuclear leukocytes and macrophages, followed by the synthesis and release of inflammatory cytokines and OS [135,164,165,166,167,168,169]. Low SS levels and venous stasis promote the pro-inflammatory processes and leukocyte infiltration (Figure 4) [99]. This initiates inflammation and cell apoptosis in the venous wall and creates conditions for increased secretion of adhesion molecules (PECAM-1, ICAM-1, and VCAM-1), which leads to a vicious cycle of alternating inflammation and remodeling of the vascular walls [170,171,172].

## 5. Discussion

Over the past 20 years, scientific research on the pathophysiology of CVD has revealed some specific molecular mechanisms and cellular interactions that have established ED as a major predisposing factor for the development of the disease [80,81]. ED is believed to precede the first clinical manifestations of CVD and represents a reliable prognostic marker for impairment of vascular homeostasis [173]. The clinical manifestation of CVD is a result of chronic inflammation of the venous wall, which increases vascular permeability and hypoxia. This is followed by the deposition of hemosiderin and the migration of leukocytes into the ECM, which favors the progression of the main CVD symptoms: pain, edema, heaviness, discomfort, itching, and discoloration in the lower extremities [174]. There are several widely accepted and generally valid theories in the scientific literature that address CVD pathophysiology: “peri-capillary fibrin cuffs” and “white cell trapping hypothesis”, which consider the role of venous reflux/obstruction, inflammation, vascular remodeling, hemodynamic changes, genetic, and social risk factors [102,175,176,177,178].

With the ED progression, vascular homeostasis is significantly impaired, ROS overproduction is promoted, vascular permeability for lipoproteins increases and the expression of inflammatory cytokines and adhesion molecules increases. Prolonged vasoconstriction is promoted, which mediates imminent, permanent cellular damage [179]. As a result of the imbalance between vasodilation and vasoconstriction of the vessel, increased levels of ROS and inflammatory mediators, deficiency of bioavailable NO, phenotypic modulation of EC, and the development of pathology occurs. Under the influence of various risk factors, ED acts as a “trigger” mechanism for changes in the morphological composition of blood vessels, which significantly increases the risk of unwanted thromboembolic events and death [180]. Chronic inflammation and abnormal ROS levels provoke oxidative damage to the endothelium due to a cycle of ROS production and chronic inflammation, which leads to endothelial injury and ultimately to endothelial dysfunction [26]. The generated by monocytes and macrophages ROS interact with NO, which causes low nitric oxide bioavailability, while simultaneously initiating the secretion of proinflammatory mediators and vascular endothelial damage, followed by vascular incompetence [84].

CVD is characterized by specific pathomorphological and hemodynamic changes in the venous vessels of the lower extremities. Venous hypertension is associated with several synergistically damaging processes that impair the functionality and efficiency of the venous system: increased hydrostatic pressure, valve dysfunction, dilation, changes in the microcirculation, hypoxia and chronic aseptic inflammation [181]. Under the influence of peripheral stretching forces, angiotensin II m type 1 receptors (AT1R) can be activated, which induce vasoconstriction and dysfunction of EC and VSMC [182]. As a result, myogenic dysregulation occurs, peripheral venous dilation is promoted, and pathophysiological conditions for the development of BB are created [99]. Increased venous pressure and venous stasis strongly affect oxygen saturation in venous vessels. The vessel walls and the innermost third of the tunica media are the most affected structures in the hypoxic environment. The physical stretching forces generated by venous hypertension compress the vasa vasorum and the venous wall, which provokes hypoxia in the tunica media and the outer part of the tunica adventitia. As a result, venous relaxation is promoted, which leads to a “vicious cycle” of alternating venous stasis and venous hypertension, deepening tissue hypoxia [131]. High plasma HIF-1α concentrations lead to increased expression of MMP-2 and MMP-9, which mediates the degradation of type III collagen and increases the synthesis of type I collagen. With the involvement of TGF-β1 and connective tissue growth factor (CTGF) in the process of EMC degradation, venous wall remodeling is begun and venous dilation is provoked. The reduced apoptotic index in these patients is associated with hypertrophy and fibrotic changes of the venous wall [183,184].

The endothelial glycocalyx is a protective structural and functional layer of membrane-bound, negatively charged proteoglycans, glycoproteins, and glycosaminoglycans, which regulates and maintains vascular homeostasis [185]. The endothelial glycocalyx represents an antithrombotic and anticoagulant surface, modulates hydrostatic and oncotic pressure between the lumen and the interstitial space, acts as a selective buffer for ions, small molecules, water, and oxygen, and represents an electrostatic barrier for cells and proteins [186]. The glycocalyx lines the luminal surfaces of endothelial cells of blood vessels and ensures the homeostatic functions of the vascular endothelium, participates in the regulation of inflammation, vascular permeability, vasodilation, modulation of coagulation and fibrinolysis, calcium dynamics, etc. [18]. An increased venous hydrostatic pressure, endothelial dysfunction, leukocyte activation, adhesion molecules secretion, decreased vascular wall shear stress, and increased ROS levels, oxidized low-density lipoprotein disrupt the glycocalyx, contributing to venous hypertension, venous dilatation, vascular dysfunction, and disease progression [187].

In patients with CVD, significant attention is paid to controlling excess weight. Endothelial damage is associated with the production of LDLox, LOX-1, and ROS. The interaction of LDLox with LOX-1 increases ROS levels, reduces NO production, and again increases LOX-1 expression, which provokes endothelial apoptosis [188]. Also, the binding of LDLox with LOX-1 activates the enzyme complex nicotinamide adenine dinucleotide phosphate (NADPH)-oxidase (the main source of ROS in EC) and leads to high levels of superoxide and the production of H_2_O_2_ [189]. Together with the formed peroxynitrite (ONOO^−^) radical, the expression of LOX-1 increases and ROS levels further increase. In addition to NADPH oxidase activation and generating superoxide, LDLox reduces NO production by competitively inhibiting eNOS and increasing the activity of the mitochondrial arginase II enzyme [190,191]. The formation of peroxynitrite (ONOO^−^) plays a significant role in endothelial dysfunction and inflammation in chronic venous diseases. Peroxynitrite is generated from the reaction between superoxide and nitric oxide. Peroxynitrite promotes oxidative modification of LDL and lipid peroxidation by forming oxLDL. Oxidative stress, activation of pro-inflammatory signaling pathways, and the release of cytokines lead to the initiation of LOX-1 upregulation, which contributes to endothelial activation, leukocyte adhesion, and vascular endothelial dysfunction, exacerbating venous wall damage [24]. Different meta-analyses, cross-sectional, and epidemiological studies reported a statistically significant positive association of overweight and obesity with the initiation and progression of CVD. Also, the overweight and obese patients have an increased risk of varicose veins, worse venous reflux, hypertension, lipodermatosclerosis, and venous ulcers, compared with normal-weight patients. Obesity was associated with higher CEAP clinical severity independent of other risk factors [192,193,194,195]. Accumulated fat (esp. visceral fat) increased intra-abdominal pressure, which causes mechanical compression of the large venous vessels, exacerbating venous stasis and promoting venous hypertension. Fat tissue releases inflammatory cytokines (TNF-α, IL-6) and leptin, initiates endothelial cell activation and leukocyte adhesion [196]. High levels of ROS, inflammatory cytokines, and increased expression of MMPs cause collagen degradation in the vessels and reduced NO bioavailability, leading to capillary leakage, microcirculatory dysfunction, venous wall remodelling, tissue fibrosis, and disease progression in obese patients, in comparison with normal-weight patients [130,138,197].

## 6. Therapeutic Aim and Strategy Outside of Mechanical Intervention

Most of the information in the scientific literature focuses on the study of hemodynamics and surgical approaches to the treatment of CVD, while the assessment of endothelial dysfunction and the associated changes in redox status and inflammation in venous diseases remains in the background. This creates a need for the introduction of an interdisciplinary approach, including panels for monitoring and evaluating biomarkers of endothelial dysfunction and oxidative stress, a review of conventional therapeutic strategies, and the introduction of new personalized medical practices for the treatment of CVD. Although compression therapy remains the gold standard in the treatment of chronic venous disease, frequent complications and relapses require a rethinking of the approach and the use of combined therapies to reduce hydrostatic pressure, oxidative stress and inflammation, and subsequent endothelial damage.

### 6.1. Venoactive Drugs with Inhibitory and Antioxidant Effects for Targeted Therapy

Venoactive drugs are pharmacologically active synthetic or natural medicinal products that are usually aimed at reducing the symptoms of CVD. Their therapeutic potential can be enhanced by combined pharmacological therapy. Management of CVD may include NF-kB inhibitors (TNF-α, IL-6, etc.) to improve the inflammatory response by reducing vascular permeability and drugs that reduce the expression of matrix metalloproteinases (MMP-1; MMP-3; MMP-9, etc.). In patients with CVD, inhibition of HIF-prolyl hydroxylase by HIF stabilizers may reduce tissue perfusion and hypoxic damage, improving tissue oxygenation in venous congestion, reducing inflammation, leading to improved vascular function. Targeted therapies to modulate shear stress with venoactive drugs such as MPFF, together with exercise regimens such as walking, may affect leukocyte–endothelial interactions, reducing leukocyte migration and OS. A wide range of natural and synthetic antioxidants can be applied to improve microcirculation, oxygenation, restore endothelial redox balance, reduce mitochondrial oxidative stress, ROS, and RNS in venous tissue. For example, alpha lipoic acid is known for its pro-inflammatory and antioxidant properties, which makes it a potential agent for the treatment of mitochondrial and endothelial dysfunction in patients with CVD. Currently, there are numerous studies that highlight the uniqueness of the redox system of oxidized-reduced forms of alpha lipoic acid (ALA). Currently, there are limited studies in the scientific literature on the positive modulation of ALA on the venous endothelium. In the context of vascular pathologies and their prevention, ALA may be an attractive redox modulator to address oxidative stress and endothelial damage.

Conventional pharmacological therapy involves systemic exposure and drug interactions of venoactive drugs, which leads to off-target and side effects, poor bioavailability, and low efficacy. A new future model in the treatment of patients with CVD is nanotherapy. It is characterized by a number of advantages over conventional pharmacological therapy, including targeted drug delivery, fewer side effects, controlled release, and the possibility of combining several venoactive drugs. Nanotherapy may be a clinically relevant and complementary approach to mechanical interventions and minimally invasive and radical surgical methods.

### 6.2. Personalized Medicine, Stratification and Assessment of the Risk of Complications

Endothelial dysfunction and oxidative stress levels can be considered as prognostic markers for vascular damage and adverse vascular events. Determining the degree of endothelial damage and measuring ROS levels can be useful in stratifying the risk of developing and progressing CVD.

### 6.3. Prevention, Diet, and Physical Activity

Despite the development of treatment strategies of vascular diseases and the increasing variety of methods for treating CVD, prevention is at most importance. Physical activity and dietary habits are the basis of measures for preventing CVD. A low-carbohydrate diet and a diet including products rich in fiber, natural antioxidants (flavonoids and polyphenols), Omega-3 polyunsaturated fatty acids (EPA and DHA), magnesium, natural redox and immunomodulators, including food supplements with redox and immunomodulatory properties, would improve the overall health of the body, thus reducing the risk of developing vascular pathologies or their recurrence. Lack of physical activity is a predisposing factor for the development of CVD in individuals with or without a genetic predisposition. Moderate physical activity, maintaining a healthy weight, and changes in eating habits will reduce inflammation and the possibility of developing the disease.

## 7. Conclusions

Recent scientific research on the pathophysiology of CVD has revealed some unambiguous molecular mechanisms and cellular interactions that have established ED as a major predisposing factor for the development of the disease. CVD is characterized by specific pathomorphological and hemodynamic changes in the venous vessels of the lower extremities, which cause a wide range of clinical symptoms with a high frequency in the general population. Hemodynamic stress is the main catalyst in the processes of EC phenotyping in patients with CVD and leads to functional incompetence of venous vessels. Inflammation, high levels of ROS and OS, and low levels of shear stress are the main causes of ECs transformation and modulation.

A more in-depth assessment and understanding of the intimate pathophysiological processes at the molecular and cellular level will allow a better analysis of the processes and factors leading to morphological and oxidative damage to venous vessels. Vascular endothelial remodeling and vascular damage often remain for a long time without the manifestation of clear clinical symptoms, which requires timely prevention, prophylaxis, and treatment of patients with CVD. It is necessary to develop more accessible screening programs and introduce new therapies to control inflammation and oxidative stress-related damage, aimed at timely intervention and prevention of chronic venous disease development and progression.

## Figures and Tables

**Figure 1 ijms-26-03660-f001:**
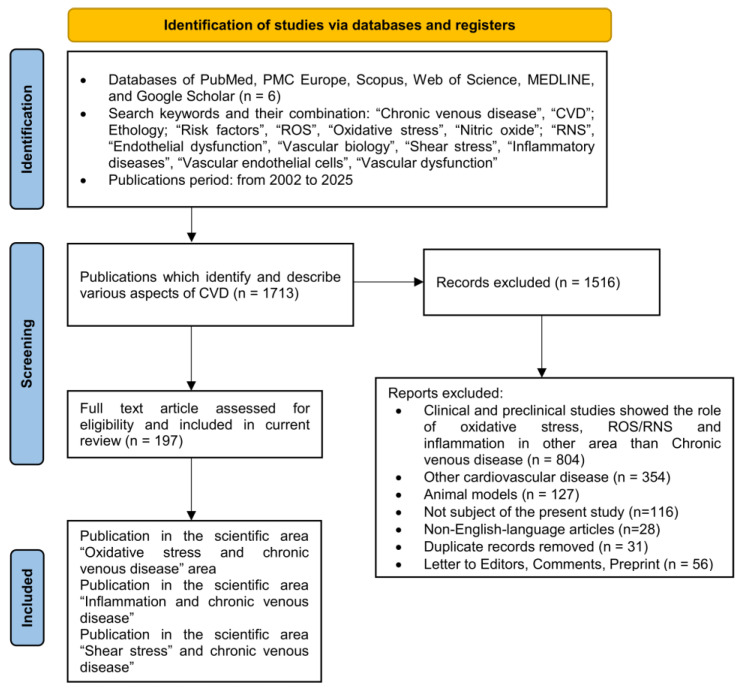
Flow diagram of databases, keywords, applicable criteria, and full-text articles assessed for eligibility and included in the current review [32].

**Figure 2 ijms-26-03660-f002:**
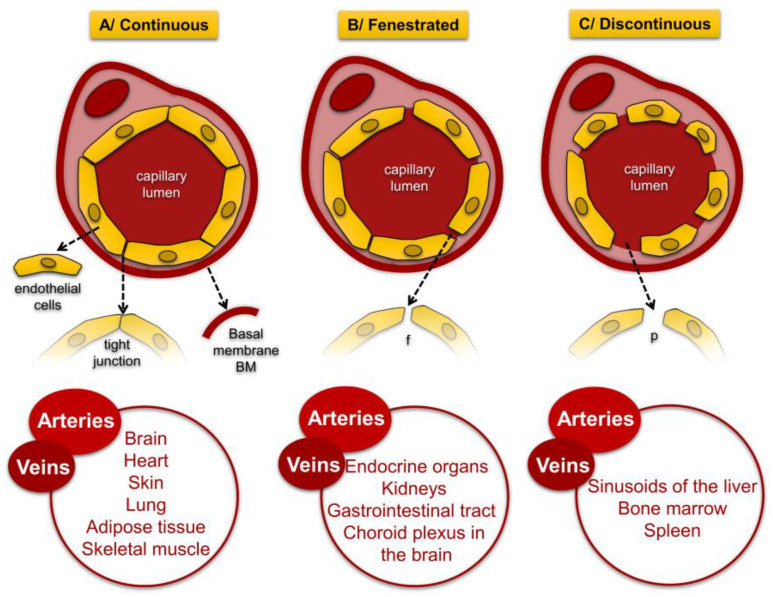
Types of endothelial layers with diameters of transcellular pores 50–300 nm. A/continuous—characterized by tight junctions that form between individual cells and high strength of attachment to the basement membrane; B/fenestrated—located close to epithelial cells, forms transcellular pores covered with plasma membrane, called “fenestras” C/discontinuous—the endothelial layer is characterized by the lowest density and the highest permeability for macromolecules [75].

**Figure 3 ijms-26-03660-f003:**
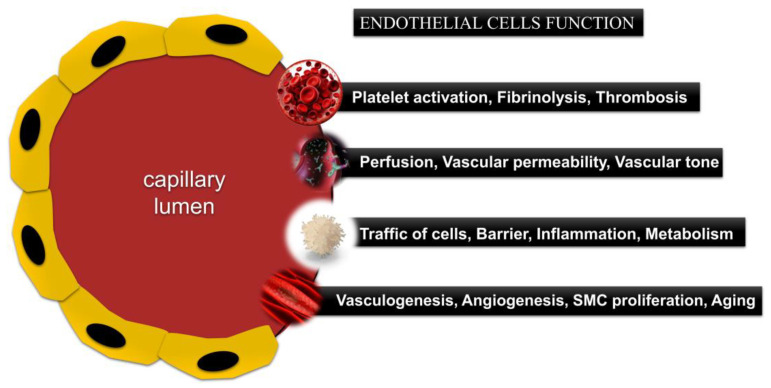
Summarized schematic representation of the main ECs processes and functions.

**Figure 4 ijms-26-03660-f004:**
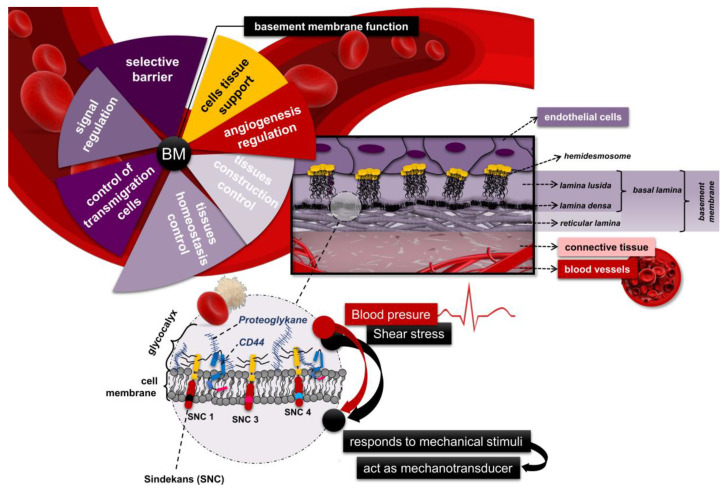
Schematic representation of the basement membrane structure and main functions, endothelial glycocalyx and shear stress.

**Table 1 ijms-26-03660-t001:** Eligible studies: Inclusion and exclusion criteria (population, study design, topic, database, search time interval outcomes etc.).

	Inclusion Criteria	Exclusion Criteria
Topic	Papers correspond to research question	Not correspond
Specify databases	PubMed, PMC Europe, Scopus, WoS, MEDLINE, Google Scholar, grey literatures	Others
Search interval	from 2002 to 2025 (with priority last 10 years)	Before 2000
Population/Target group	Studies in humans/patients, cell lines, biological samples	Animal studies
Study design (papers)	Scoping, Narrative, Systematic Review, Meta-Analysis, Case Report, Full articles	Others
Study design (language)	Original studies in English	Others
Accessed review protocol	PRISMA, PICOS, Cochrane Handbook, etc.	Others
Prospective Register	PROSPERO protocols	Others
Outcomes	Outcomes papers that provide 1/an unbiased and exhaustive overview and 2/summarize the current scientific data on the role of vascular endothelial dysfunction in chronic venous disease, associated with inflammation, oxidative stress, and shear stress	Others
Declare competing interests	The authors declare no competing interests	Others

## Data Availability

The data presented in this study are available on request from the corresponding author.

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
