# Peer review of "A Systematic Review of Endothelial Dysfunction in Chronic Venous Disease—Inflammation, Oxidative Stress, and Shear Stress"

_ijms, 2025, doi:10.3390/ijms26083660_

Round 1
Reviewer 1 Report
Comments and Suggestions for Authors
Dear Authors,
Overall I consider this manuscript to be an interesting paper. However, I have several comments and suggestions, in order to improve it. First of all, this paper looks more like a narrative review for me, so I suggest you to revise the manuscript accordingly.
Major concerns:
- Abstract: Considering the article type - review, several data in this regard should be mentioned in the abstract: how many papers were searched? how many articles were included in the literature corpus? which data-bases were accesed? etc.
- Introduction, line 50: ...`obesity (BMI> 25 kg/m2)`. This is wrong, because considering the BMI values, >25kg/m2 is overveight (https://www.ncbi.nlm.nih.gov/books/NBK535456/), >30kg/m2 is obese (https://pubmed.ncbi.nlm.nih.gov/39595090/). Additionally, inflammation and obesity interplay plays a key role in CVD progression, leading to advanced endothelial dysfunction, aspect that should be discussed.
- Methods section: The research question(s) should be celarly stated.
- Methods section: What about risk of bias and quality assessment?
- Methods, line 100 - Inclusion Criteria: `The number of identified original articles and studies presenting the role of free radical damage, oxidative stress and endothelial dysfunction in various cardiovascular diseases exceeded 4500.` - What does this have to do with inclusion criteria? The inclusion criteria are poorly described, and a `Data Collection Process` subsection should be introduced in the Methods section.
- Methods, line 108. The exclusion criteria are poorly described. This section should be extensively revised.
- Results section: in a systematic review, the results should be presentaed more...systematically, not combined with discussion. Blending results and discussion is more suitable for a narrative review.
- Discussion section: Glycocalyx disruption, endothelial dysfunction and vascular remodeling ar underlying mechanisms and treatment targets of CVD. This aspect should be introduced in your discussion (https://pubmed.ncbi.nlm.nih.gov/39873224/, https://pubmed.ncbi.nlm.nih.gov/36810649/, https://doi.org/10.1007/s12325-023-02657-0., https://pubmed.ncbi.nlm.nih.gov/36315163/, etc.)
Minor concerns:
- Figure 1: `Records excluded** (n=1336)`. What does the double asterix (`**`) mean? It should be stated in figure caption.
- Lines 101-103 should be moved to `Data Collection Process` subsection.
- I recommend you to consider the above suggested literature, as well other valuable scientific references.

Author Response
RESPONSES TO REVIEWER 1
We appreciate the reviewers’ comments.
Dear Reviewer,
Thank you very much that you help us to improve our manuscript. All changes in the text are in red color.
Major concerns according to Reviewer’s recommendations
Point 1. Abstract: Considering the article type - review, several data in this regard should be mentioned in the abstract: how many papers were searched? how many articles were included in the literature corpus? which data-bases were accesed? etc.
The answer to Point 1: Done
Point 2. Introduction, line 50: ...`obesity (BMI> 25 kg/m2)`. This is wrong, because considering the BMI values, >25kg/m2 is overveight (https://www.ncbi.nlm.nih.gov/books/NBK535456/), >30kg/m2 is obese (https://pubmed.ncbi.nlm.nih.gov/39595090/). Additionally, inflammation and obesity interplay plays a key role in CVD progression, leading to advanced endothelial dysfunction, aspect that should be discussed.
The answer to Point 2: Done (discussion and references section)
Point 3. Methods section: The research question(s) should be celarly stated
The answer to Point 3: Done
Point 4. Methods section: What about risk of bias and quality assessment?
The answer to Point 4: Done
Point 5. Methods, line 100 - Inclusion Criteria: `The number of identified original articles and studies presenting the role of free radical damage, oxidative stress and endothelial dysfunction in various cardiovascular diseases exceeded 4500.` - What does this have to do with inclusion criteria? The inclusion criteria are poorly described, and a `Data Collection Process` subsection should be introduced in the Methods section.
The answer to Point 5: Done
Point 6. Methods, line 108. The exclusion criteria are poorly described. This section should be extensively revised.
The answer to Point 6: Done
Point 7. Results section: in a systematic review, the results should be presentaed more...systematically, not combined with discussion. Blending results and discussion is more suitable for a narrative review.
The answer to Point 7: Done
Point 8. Discussion section: Glycocalyx disruption, endothelial dysfunction and vascular remodeling ar underlying mechanisms and treatment targets of CVD. This aspect should be introduced in your discussion (https://pubmed.ncbi.nlm.nih.gov/39873224/, https://pubmed.ncbi.nlm.nih.gov/36810649/, https://doi.org/10.1007/s12325-023-02657-0., https://pubmed.ncbi.nlm.nih.gov/36315163/, etc.)
The answer to Point 8: Done (discussion and references section)
Minor concerns according to Reviewer’s recommendations
Point 1. Figure 1: `Records excluded** (n=1336)`. What does the double asterix (`**`) mean? It should be stated in figure caption.
The answer to Point 1: Corrected
Point 2. Lines 101-103 should be moved to `Data Collection Process` subsection.
The answer to Point 2: Done
Point 3. I recommend you to consider the above suggested literature, as well other valuable scientific references.
The answer to Point 3: Done
Sincerely yours
Prof. Ekaterina Georgieva, Ph.D
Department of General and clinical pathology, forensic medicine, deontology and dermatovenerology, Faculty of Medicine, Trakia University, Stara Zagora, 6000 Bulgaria
Reviewer 2 Report
Comments and Suggestions for Authors
A review titled “A systematic review of Endothelial Dysfunction in Chronic Venous Disease – inflammation, oxidative stress, and shear stress” is submitted for a potential publication. This reviewer has some concerns that are mentioned below-
- Authors should be aware of the term “CVD”, which is already known as cardiovascular disease and avoid using it for chronic venous disease.
- The significance of writing this review is not clear. What is the importance of this review? How would it benefit the scientific community?
- This review doesn’t seem to provide any significant research directions but provides more general description about the role of different contributing factors, with structure & function of endothelium, most of which are well known in the literature.

Author Response
RESPONSES TO THE REVIEWER 2
We appreciate the reviewers’ comments.
Dear Reviewer,
Thank you very much that you help us to improve our manuscript. All changes in the text are in blue color.
Point 1. Authors should be aware of the term “CVD”, which is already known as cardiovascular disease, and avoid using it for chronic venous disease.
The answer to Point 1: The abbreviation "CVD" is accepted for use and can refer to both "chronic venous disease" and "cardiovascular disease", and the use depends on the medical context. The links provided are a small part of the many examples.
https://www.jvsvenous.org/article/S2213-333X(23)00322-0/fulltext
https://www.mdpi.com/2077-0383/10/15/3239
https://www.jvsvenous.org/article/S2213-333X(24)00308-1/fulltext
https://www.jvsvenous.org/article/S2213-333X(21)00431-5/fulltext
https://www.jvsvenous.org/article/S2213-333X(24)00152-5/fulltext
https://www.mdpi.com/2077-0383/10/15/3239
https://www.mdpi.com/1422-0067/24/3/1928
https://www.mdpi.com/1648-9144/59/6/1034
https://academic.oup.com/ced/article-abstract/47/7/1228/6693009#google_vignette
etc.
CVD is abbreviation for chronic venous disease and outlined in the guidelines and protocols for management of the chronic venous disease and complications (and their update), according to clinical practice guidelines of "Society for Vascular Surgery", "American Venous Forum", "American Vein and Lymphatic Society" and "European Society for Vascular Surgery" and as follows:
https://www.jvsvenous.org/article/S2213-333X(23)00322-0/fulltext
https://www.ejves.com/article/S1078-5884(21)00979-5/fulltext
https://www.ejves.com/article/S1078-5884(22)00024-7/fulltext
At the moment, we cannot use another abbreviation for chronic venous disease except CVD until a new abbreviation is introduced in the medical and scientific societies and practice. The specific condition is usually clarified in medical discussions and scientific articles, and this practice aims to differentiate and avoid confusion in medical conditions. According to the general rules and information, in the current manuscript, we have explicitly indicated that "CVD" is chronic venous disease.
Point 2. The significance of writing this review is not clear. What is the importance of this review? How would it benefit the scientific community?
Answer 2: We corrected
Point 3. This review doesn’t seem to provide any significant research directions but provides more general description about the role of different contributing factors, with structure & function of endothelium, most of which are well known in the literature.
Answer 3: We corrected and attach new inforamtion
Sincerely yours
Prof. Ekaterina Georgieva, Ph.D
Department of General and clinical pathology, forensic medicine, deontology and dermatovenerology, Faculty of Medicine, Trakia University, Stara Zagora, 6000 Bulgaria
Round 2
Reviewer 1 Report
Comments and Suggestions for Authors
The manuscript is significantly improved, being now clearer for the reader and more appropriate considering the scientific rigour. Congratulations!
Author Response
Dear Reviewer,
Thank you for your assistance in improving our manuscript.
Reviewer 2 Report
Comments and Suggestions for Authors
This review is improved.
Author Response

(The authors gave the same response as above.)
